# Electrochemical and Surface Analytical Study on the Role of Poly(butylene-succinate)-L-proline during Corrosion of Mild Steel in 1 M HCl

**George M. Tsoeunyane *** and **Elizabeth M. Makhatha**

Department of Metallurgy, Faculty of Engineering and Built Environment, DFC Campus 1,
University of Johannesburg, Johannesburg 2094, South Africa; emakhatha@uj.ac.za

* Correspondence: m.tsoeunyane.mg@gmail.com; Tel.: +27-11559-6194

**Abstract:** The synthesis and corrosion inhibition performance of poly(butylene-succinate)-L-proline (PBSLP) prepared by solution polymerization are reported. PBSLP was characterized by FTIR, XRD, and SEM/energy dispersive X-ray (EDX). PBSLP was used to protect mild steel in 1 M hydrochloric acid. An SEM and an atomic force microscope (AFM) were used to characterize the surface morphology of the mild steel coupons. Potentiodynamic polarization and electrochemical impedance spectroscopy (EIS) were used to characterize the inhibition mechanism of PBSLP, and the inhibitor was a mixed-type corrosion inhibitor with a maximum corrosion inhibition efficiency of 93.0%. Adsorption studies revealed the adsorption of PBSLP to be a monolayer process and therefore, obeyed the Langmuir isotherm model.

**Keywords:** potentiodynamic polarization; electrochemical impedance spectroscopy; inhibition mechanism; surface morphology; mild steel; Langmuir adsorption isotherm

## 1. Introduction

Mild steel is an alloy which finds wide application in many industrial fields because of its excellent mechanical properties, low cost, and easy workability. However, mild steel exhibits poor corrosion resistance which results in the huge loss of metal [1,2]. Among all corrosion prevention measures, corrosion inhibitors carry advantages of economy, high-efficiency, and facile feasibility [3–8]. The use of corrosion inhibitors is a very effective and economic way of protecting metals in industrial processes such as pickling, industrial acid cleaning, and oil and gas well acidizing [9–11].

Hydrochloric acid is the most used acid because it is economic and comparatively causes fewer issues as compared to other mineral acids. The ability of metal chloride formation by hydrochloric acid serves as an advantage for the wide usage of the acid because of high solubility as compared to nitrates and sulphates [12]. The high solubility causes a lower polarizing effect and does not affect the corrosion process during acid cleaning and pickling processes.

Designing nontoxic corrosion inhibitors has been a challenge because of the complex nature of adsorption of corrosion inhibitors to the metal surface, which is not well understood [13]. Defining the molecular structure of corrosion inhibitors would lead to the ultimate explanation of the inhibition mechanisms. Small changes in molecular structure can dramatically change the surface chemistry between the metal surface and the inhibitor. The molecules with heteroatoms such N, O, and S are reported to have a good inhibitive effect because of the lone pair of electrons they have [14]. The efficiency of the corrosion inhibitors to protect metals against corrosion is connected to the barrier that is formed between the metal solution interface through adsorption [5,15,16].

Many corrosion inhibitors have been reported, and some inhibitors are characterized by low inhibition efficiency, toxicity, difficulty in handling, and incurring costs of preparation [17]. Therefore,

we hereby report a polymer composite that is biodegradable, non-toxic, and easy to prepare and handle as a corrosion inhibitor of mild steel. The method of polymer composite preparation was similar to the one that was reported in our previous publication [18]. The inhibitor would slow down surface degradation of mild steel when exposed to corrosive media. The source of corrosive media used in this study was 1 M hydrochloric acid. The data (supplementary information) used in this study have been provided by the link at end of the manuscript.

## 2. Materials and Methods

### 2.1. Materials and Chemicals

All chemicals used in this study were analytical grade, and they were used without further purification. Mild steel had the chemical composition of Mn—0.249; C—0.061; P—0.015; Cr—0.027; S—<0.1; Ni—0.016; Si—0.013; Mo—0.024; Al—0.040; Co—0.011; Mg—0.013; Se—0.039; Zn—0.012; the balance was Fe (wt %t). The chemical composition was obtained with an arc-spark optical emission spectrometer.

### 2.2. Mild Steel Coupons Preparation

The mild steel coupons were cut into $5 \times 2 \times 0.5$ cm dimensions; a 3 mm hole was drilled towards the edge of each coupon. The sectioned samples were ground with emery paper, grit (200, 600, 800, and 1200) starting with the coarse and finishing with the fine paper until a mirror finish-appearance was obtained. The coupons were rinsed with distilled water and degreased with acetone. They were dried in an oven at 50 °C and then, stored in a desiccator, and they were ready to be used.

### 2.3. Electrochemical Studies

Potentiodynamic polarization tests were performed on the flat mild steel coupon; an area of $12.54 \, \text{cm}^2$ was exposed in 250.0 mL corrosive media. The coupons were used as a working electrode in the flat corrosion cell. The platinum gauze meshed wire was used as the counter electrode, while a saturated calomel electrode (SCE) was used as a reference electrode. The potential measurements were performed in the range of $-2.5$ to $+2.5$ V with a scan rate of 1.0 mV/s. The corrosion potential *(Ecorr)*, corrosion current density *(Icorr)*, and Tafel slopes were determined from the tangent extrapolation on the polarization curves.

It is important to remember that potential scan rate has an important role in minimizing the effects of distortion in Tafel slopes and corrosion current density analyses, as widely and previously reported [19–22]. However, based on these reports, the adopted 1 mV/s can be considered without deleterious effects on the Tafel extrapolation method to determine the corrosion current densities of the examined samples. The corrosion potential *(Ecorr)*, corrosion current density *(Icorr)*, and Tafel slopes were determined from the tangent extrapolation on the polarization curves.

Electrochemical impedance spectroscopy (EIS) measurements were performed from a high frequency of 20 kHz to a low frequency of 0.1 Hz, in the potential range of $-2.5$ to $+2.5$ V. The sinusoidal potential of 5.0 mV was applied to the system. The experiments were performed in fast mode. The Z fit tool from the EC-Lab software was used to analyze the EIS data. The Nyquist and Bode plots were used to explain reaction mechanisms and experimental results. The EIS parameters, such as solution resistance (Rs), charge transfer resistance (Rct), and constant phase element (CPE), were obtained from the Nyquist plot. The equivalent electrical circuit (EEC) was proposed and modelled from the experimental data.

### 2.4. Surface Analysis

The surface morphology and topography of inhibited and uninhibited mild steel coupons were studied with a TESCAN scanning electron microscope (SEM) and an atomic force microscope (AFM). The micrographs and 3D plot micrographs were obtained indicating the roughness of mild steel

coupons immersed in different corrosive media. The morphological appearance on the mild steel coupons was explained with reference to the unexposed coupon (blank). Moreover, Gwyddion software was used to estimate the average surface roughness of the coupons.

$$IE_p = \left( \frac{i'_{corr} - i_{corr}}{i'_{corr}} \right) \times 100 \tag{1}$$

where $IEp$, $i'_{corr}$, and $i_{corr}$ were corrosion inhibition efficiency, and corrosion current density without and with poly(butylene-succinate)-L-proline (PBSLP), respectively.

$$IE_{EIS} = \left( \frac{R'_{ct} - R_{ct}}{R'_{ct}} \right) \times 100 \tag{2}$$

where $IE_{EIS}$, $R'_{ct}$, and $R_{ct}$ were corrosion inhibition efficiency, and charge transfer resistance without and with PSBLP, respectively.

$$Q = \frac{1}{(jW)^a Y_o} \tag{3}$$

where $\alpha$ can be extracted from the imaginary impedance frequency graph, $w$ is the angular frequency, $\alpha$ is a constant phase exponent, $j$ imaginary current, and $Y_o$ is a function of $\alpha$ and impedance. The parameters $\alpha$ and $Y_o$ are expressed in Equations (4) and (5), respectively.

$$a = \frac{dlog(-Z_j)}{dlog(f)} \tag{4}$$

where $Z$ is an impedance and $f$ is frequency.

$$Y_o = sin\left( \frac{a\pi}{2} \right) \frac{-1}{Z_j(f)(2\pi f)^a} \tag{5}$$

$$\frac{C}{\theta} = \frac{1}{k} + C \tag{6}$$

where $C$ is the concentration of the inhibitor, $k$ is adsorption equilibrium constant, and $\theta$ surface coverage.

$$log\theta = logK_F + \frac{1}{n}logC \tag{7}$$

where $K_F$ is the adsorption capacity and $\frac{1}{n}$ is the adsorption capacity. It indicates the relative distribution of energy and the heterogeneity of the adsorbate sites.

$$\theta = \left( \frac{i'_{corr} - i_{corr}}{i'_{corr}} \right) \tag{8}$$

where $\theta$ is the fraction surface coverage.

## 3. Results and Discussion

### 3.1. Characterization of PBSLP

#### 3.1.1. FTIR

The method for PBSLP inhibitor preparation was adopted from R. Karthikaiselvi et al. [23]. Fourier Transform Infrared (FTIR) spectroscopy was used to characterize chemicals used in polymer matrix preparation and the prepared matrix. Figure 1 shows the FTIR spectra lines of PBS (red) and proline (black) molecular structures shown in Figures 2 and 3, respectively, and PBSLP (blue). All materials used for FTIR characterization were powder. A Nicolet iS50 FTIR spectrometer equipped

with a diamond crystal beam splitter was used, where 1.0 g of the sample was placed on the stage and tightly sealed to resist ambient humidity interference. The samples were analyzed in the spectral range of 500 to 4000 cm$^{-1}$.

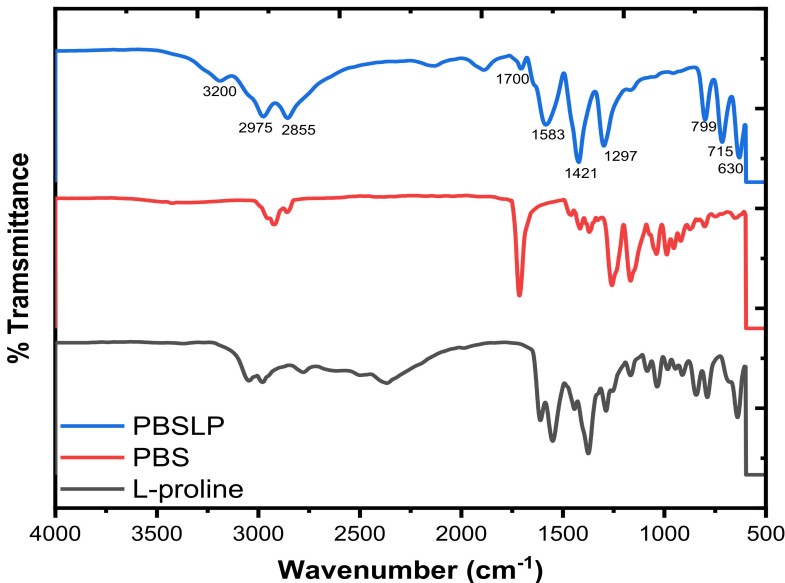

**Figure 1.** FTIR spectra of L-proline, polybutylene(succinate) (PBS), and poly(butylene-succinate)-L-proline (PBSLP).

**Figure 2.** Poly(butylene-succinate).

**Figure 3.** L-proline.

The spectral line of the polymer composite is different from the spectral lines of PBS and L-proline; this is because of the rearrangement of atoms in the formation of the composite. In the PBSLP spectrum, the presence of N-H is observed at 3200 cm$^{-1}$ as a symmetric stretch, and its distortion is observed at 1583 cm$^{-1}$. The C=O transition band is observed at 1700 cm$^{-1}$ with a weak transition peak as opposed to a normal strong transition peak. The change is brought by the hydrogen bonding which took place between N-H of proline and the carbonyl group of PBS. This hydrogen bonding resulted in a pseudo-grafted polymer composite.

As expected, the presence of C-H and COO transition bands was observed in all three materials. Despite occurring in all materials, their wavenumbers (energy) appearance of the transitions shifted

in magnitudes. The electron density within the molecules affects the vibration bands induced by infrared radiation. The spatial orientation has a similar effect to electron density. In our preparation, the semi-polymerized L-proline is grafted to polybutylene succinate (PBS) C=O hydrogen bonding of N-H. Therefore, this increases electron density within the macromolecule; hence, different transitions of similar bonds are observed in L-proline, PBS, and PBSLP, respectively.

In solution polymerization, the by-product is a water molecule. Solution polymerization is an exothermic process. The O-H characteristics were observed in L-proline, which, due to stereochemistry overlap between N-H and C-H transition bands (Figure 1). The OH and H from the amide group were the active centers of polymerization; their reaction resulted in $H_2O$ as a by-product of semi-polymerization of proline. The polymerization of proline is terminated by keeping the solution media in the cold environment. The reaction between semi-polymerized proline and the PBS matrix is initiated by raising the pH of the media to basic conditions 9–10 in the cold environment. As explained, the reaction happens through hydrogen bonding.

### 3.1.2. XRD and SEM–EDX

A Rigaku Ultima IV X-ray diffraction spectrometer was used to characterize PBS and PBSLP inhibitor. Of each material, 2.0 g was placed on a glass slide and pressed to obtain a flat surface. The glass slide was placed on a single-stage sample holder and the sample was scanned from a θ degree angle of 5 to 90 at 1.0 degree/min with an X-ray of 40 KV. The K-beta filter was used to reduce the intensity of the wavelength and a step-width of 0.05 degrees was applied. The diffractograms of PBS and PBSLP are shown in Figure 4a,b, respectively. The intensity of the peaks in PBSLP is 10 times in magnitude as compared to PBS. The indication of this observation is the semi-crystallinity of PBSLP, while PBS is relatively amorphous.

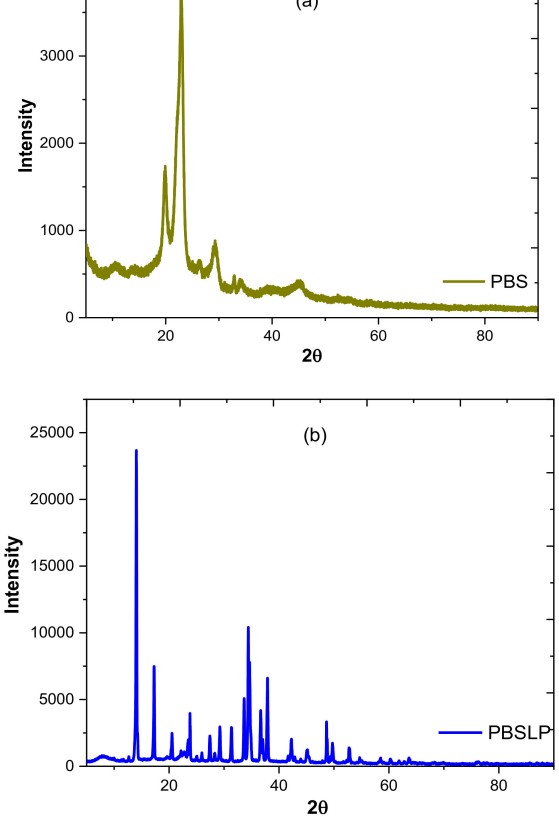

**Figure 4.** Diffractogram of (**a**) PBS and (**b**) PBSLP.

The morphological appearance of the molecules was further investigated with SEM (Figure 5) and their elemental composition with energy dispersive X-ray (EDX) (Figure 6). The SEM micrographs show the continuous phase of PBS (Figure 5a), while needle-like structures are observed for PBSLP in Figure 5c. A secondary electron (SE) detector was used to obtain the micrographs at a voltage of 20 kV. The samples were initially coated with graphite to increase their conductivity. A PBSLP EDX spectrum (Figure 6c) shows the presence of nitrogen (N) atom, while the PBS EDX spectrum (Figure 6a) shows the absence of an N atom. The N atom was introduced by proline to the PBS polymer by solution polymerization. The N and O oxygen atoms present within the PBSLP inhibitor increase the activity of the molecule as an inhibitor.

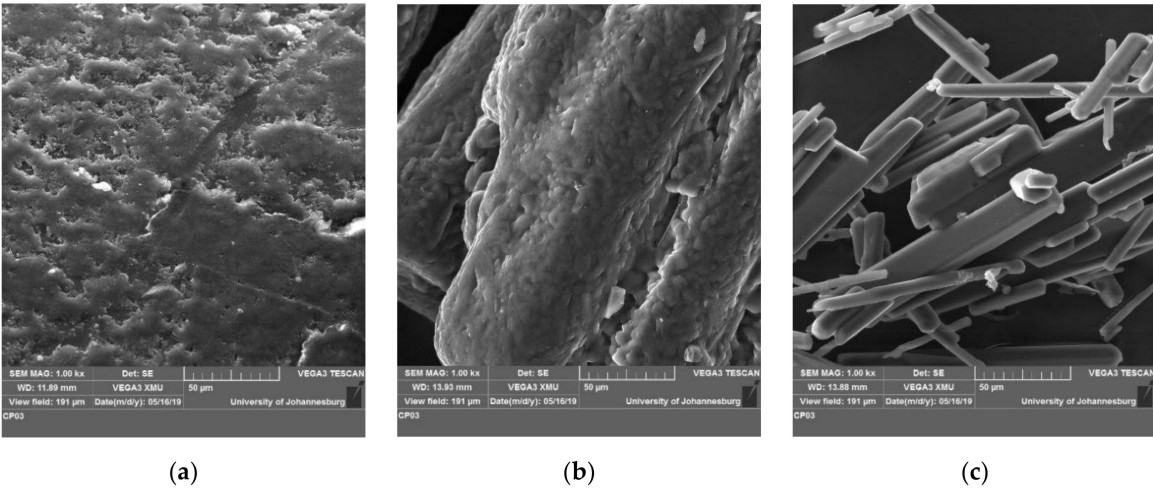

| (**a**) | (**b**) | (**c**) |

**Figure 5.** SEM micrographs for (**a**) PBS, (**b**) L-proline and (**c**) PBSLP.

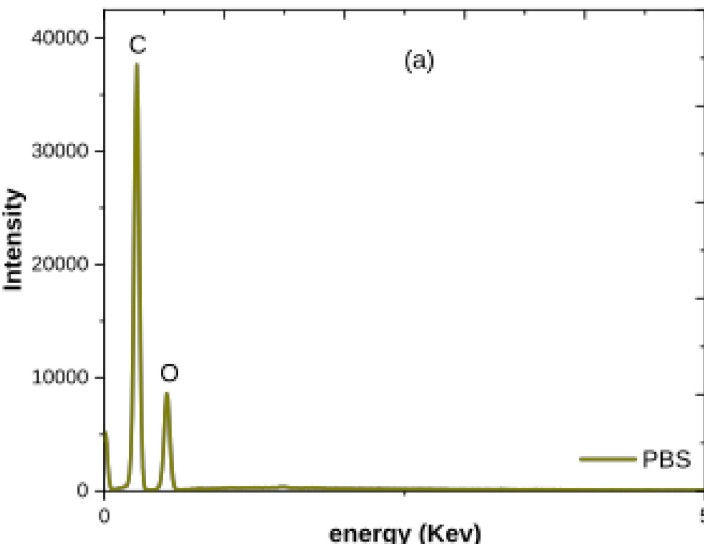

**Figure 6.** *Cont.*

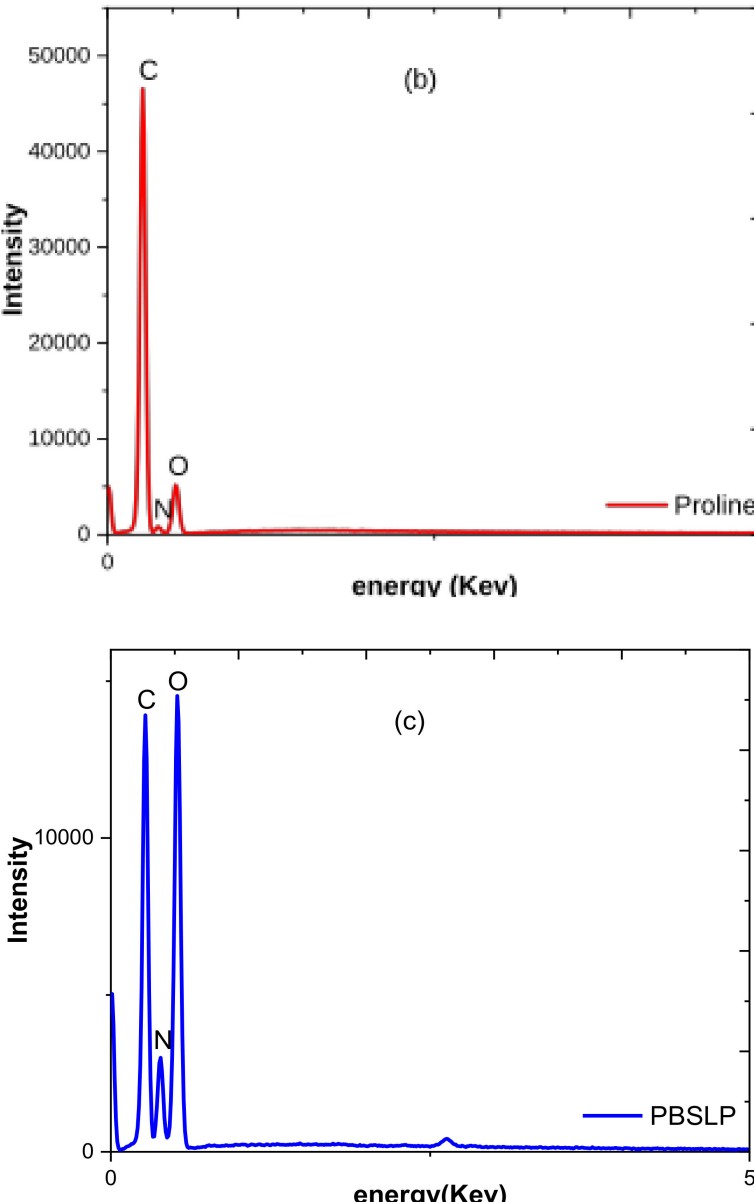

**Figure 6.** Energy dispersive X-ray (EDX) graphs showing the excitation of different elements within (**a**) PBS, (**b**) L-proline, and (**c**) PBSLP.

## 3.2. Electrochemical Analysis

### 3.2.1. Potential vs. Time

The variation of potential, *E*, as the function time of mild steel in 1 M HCl with and without the presence of PBSLP as different concentrations is shown in Figure 7. The immediate potential drop is observed for the blank solution, and the potential moved towards the negative side. The implication is that mild steel was being activated to corrosion. In the concentration range from 200 to 400 ppm, initially, the potential moved towards the positive side and eventually changed direction towards the negative side before the potential was able to stabilize.

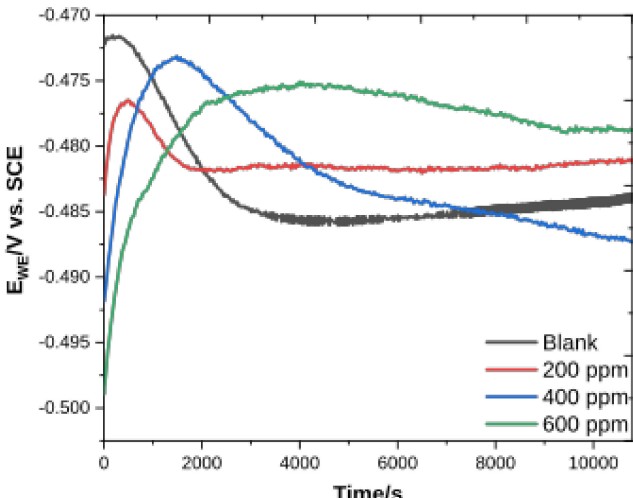

**Figure 7.** Open circuit potential (OCP) of mild steel immersed in 1 M HCl at different concentration for 3 h.

The behavior observed could be explained in terms of the protective film that was formed on mild steel, which appeared in the form of passivation. Due to the insufficient number of PBSLP macromolecules to protect mild steel, the potential suddenly changed, showing the mild steel's activation to corrosion. Different behavior was observed at 600 ppm concentration of PBSLP. At the start of the experiment, the potential moved in the positive side until the potential stabilized without showing any signs of activation. Therefore, 600 ppm was the indication of enough concentration of PBSLP to protect mild steel against corrosion. The OCP was performed for 3 h for each test.

### 3.2.2. Potentiodynamic Polarization

Electrochemical methods are an effective way of studying accelerated corrosion of materials. The corrosion behavior of mild steel coupons in 1 M HCl without and with different concentrations of PBSLP is represented in terms of Tafel plots in Figure 8. The potential scans were performed in a positive direction. In the concentration range of 0 (blank) to 400 ppm, the current on the cathodic branch showed a rapid increase in the current at the beginning of the scans and then, dropped. Different behavior was observed at 600 ppm, where the current did not increase at the beginning of the experiment.

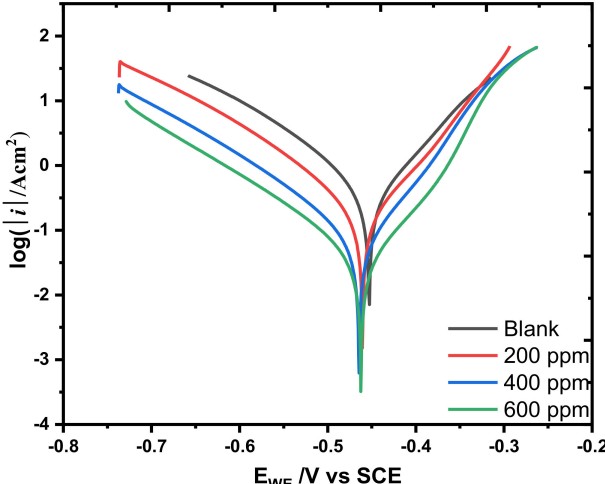

**Figure 8.** Tafel plots of mild steel in 1 M HCl at different concentrations of PBSLP at a scan rate of 1.0 mV/s.

The behavior observed implied that the more macromolecules present in the solution, the better the protection of the mild steel that would be achieved. Furthermore, the results showed that the rate at which PBSLP macromolecules adsorbed onto the mild steel also depended on the concentration. The reaction on the cathodic side is due to the reduction in hydrogen ions (see Equation (8)). The PSBLP inhibitor blocked the active sides, and the corrosion rate of the mild steel coupons was decreased. The anodic branches of the concentration from 200 to 600 ppm are affected as compared to the anodic branch of the blank solution. Thus, the inhibitory action of the PBSLP inhibitor affects both the anodic and cathodic side. On the anodic side, the reaction was due to the dissolution of Fe from mild steel (see Equation (2)).

$$2H^+ + 2e^- \rightarrow H_2 \text{ cathodic reaction} \tag{9}$$

$$Fe \rightarrow Fe^{2+} + 2e^- \text{ anodic reaction} \tag{10}$$

The evidence of PBSLP affecting both cathodic and anodic sides is represented in Table 1. The cathodic and anodic slopes of the blank solution are significantly higher than the slopes of the inhibitor concentrations.

**Table 1.** Potentiodynamic polarization parameters of mild steel in 1 M HCl in the presence and absence of PBSLP.

| Conc (ppm) | $E_{corr}$/mV | $i_{corr}$/Acm$^{-2}$ | $\beta_A$/(mV/Decade) | $\beta_C$/(mV/Decade) | CR/mmpy | %IE | θ |
|---|---|---|---|---|---|---|---|
| Blank | −453.11 | 0.232 | 65.8 | 68.3 | 0.212 | - | - |
| 200 | −460.72 | 0.055 | 55.9 | 61.3 | 0.0502 | 76.34 | 0.766 |
| 400 | −463.93 | 0.037 | 50.6 | 62.4 | 0.0340 | 83.98 | 0.839 |
| 600 | −461.96 | 0.016 | 48.3 | 53.2 | 0.0150 | 92.94 | 0.929 |

Moreover, the difference between the corrosion potential of the blank solution and different concentrations is less than 85.0 mV. Therefore, the PBSLP composite was a mixed-type corrosion inhibitor [24,25]. The Tafel plots in the presence of PBSLP shifted towards the negative direction. The mixed-type corrosion inhibitors block the active sides of both the cathodic and anodic reactions. In this study, the anodic reaction was due to the oxidation of Fe from the mild steel surface.

The results indicated that the addition of PBSLP to corrosive media reduced the corrosion rate of mild steel and the inhibition efficiency percentage was increased (see Table 1). The maximum inhibition efficiency percentage of 92% was obtained and the minimum corrosion rate was 0.0150 mmpy. The inhibition percentages obtained from the electrochemical reactions are based on the charge transfer. The aspect of mass transfer is limited; that is the reason high inhibition efficiencies are obtained with electrochemical studies as compared to gravimetric analysis. The corrosion inhibition efficiencies were calculated from the current densities by using Equation (1).

### 3.2.3. Electrochemical Impedance Spectroscopy (EIS)

As a non-destructive method, EIS was used to study and monitor the adsorption of PBSLP onto the mild steel's surface. EIS explains the electrode-electrolyte interface, electrochemical reactions by electrical circuit analogues in the corrosion system. The Z-fit tool on EC-Lab software was used to analyze the impedance data.

The results of EIS measurements are represented in Figure 9 as a Nyquist plot and Figure 10 as a Bode plot. The EIS parameters are represented in Table 2. The explanation of EIS data involves the use of equivalent electrical circuits (EEC); Figure 11 represents the EEC that was used to explain the impedance data of the inhibition of mild steel by the PBSLP inhibitor in 1 M HCl. The ECC models the solution–metal interface. The corrosion inhibition efficiencies were calculated from the charge transfer resistance ($R_{ct}$) by Equation (2).

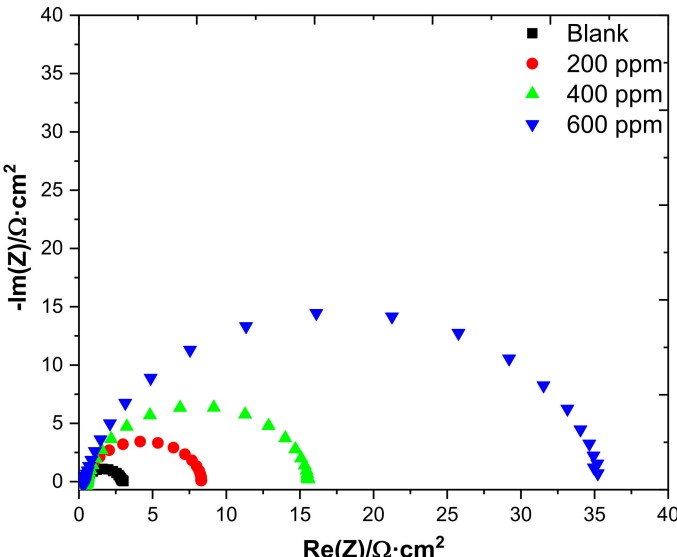

**Figure 9.** Nyquist plot of mild steel in 1 M HCl at different concentrations of PBSLP with a sinusoidal potential of 5 mV.

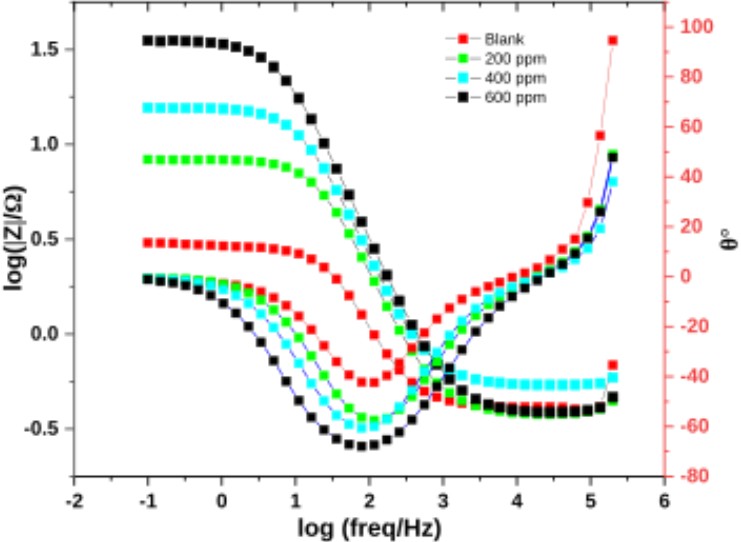

**Figure 10.** Bode plot of mild steel in 1 M HCl at different concentrations of PBSLP with the sinusoidal potential of 5 mV.

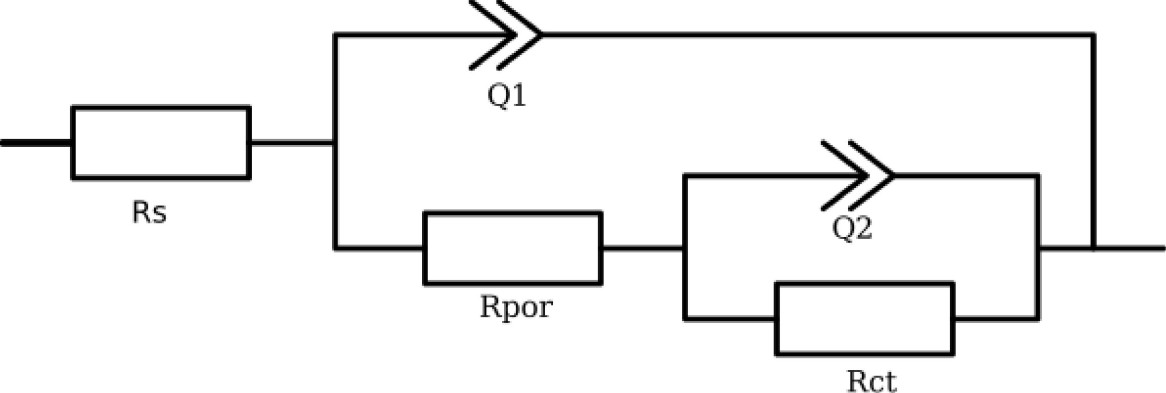

**Figure 11.** Electrical circuit model for the impedance of PBSLP adsorbed onto the mild steel surface.

**Table 2.** The EIS parameters of mild steel in 1 M HCl in the absence and presence of PBSLP.

| Conc/ppm | $R_s/\Omega$ | $Q_1/F.s\char`^(\alpha-1)$ | $\alpha_1$ | $R_{por}/\Omega$ | $Q_2/F.s\char`^(\alpha-1)$ | $\alpha_2$ | $R_{ct}/\Omega$ | %IE | $\chi^2$ |
|---|---|---|---|---|---|---|---|---|---|
| **Blank** | 0.4131 | $3.231 \times 10^{-3}$ | 0.901 | 0.088 | $5.692 \times 10^{-5}$ | 1 | 2.058 | - | 0.045 |
| **200** | 0.4785 | $1.971 \times 10^{-3}$ | 0.894 | 0.154 | $0.5608 \times 10^{-6}$ | 0.895 | 4.833 | 57 | 0.059 |
| **400** | 0.4343 | $1.251 \times 10^{-3}$ | 0.898 | 0.267 | $1.251 \times 10^{-7}$ | 0.898 | 15.13 | 86 | 0.024 |
| **600** | 0.3808 | $1.083 \times 10^{-3}$ | 0.882 | 4.34 | $6.493 \times 10^{-9}$ | 0.925 | 30.75 | 93 | 0.044 |

As observed in Figure 9, the increase in inhibitor concentration increases the radius of the depressed semicircles of the Nyquist plot. The increase in size is due to high charge transfer resistance at the metal–solution interface that results from film formation by the PBSLP inhibitor. The maximum inhibition efficiency obtained was 93%, as shown in Table 2. The film formed between the metal–solution interface was determined to have capacitive behavior but due to the imperfection of the film formed, the double-layer capacitor was modelled by a constant phase element (Q) in the EEC modelling [26], shown in Figure 11. The $R_{ct}$ values increase with the increasing PBSLP concentration; this is true with $R_{por}$ values.

The constant phase element is expressed in Equation (3). As the indication of the protection of mild steel by the PBSLP inhibitor, the real and imaginary parts where the inhibitor was applied are higher in magnitude as compared to the blank solution. The exact time constant indicating the number at which the magnitude has been increased between the blank solution and where the inhibitor was applied cannot be easily seen in the complex impedance plane [27]. Therefore, the sizes of the depressed semicircles are used to explain the impedance of mild steel dissolution in the presence of the inhibitor.

It is remarked that CPE characterizes the impedance of a phase element, which is described by $CPE = [C(jw)^\alpha]^{-1}$, where C is the capacitance; j is the current (imaginary number: $-1^{0.5}$); w is the angular frequency; and $-1 < \alpha < 1$. When CPE attains $\alpha = 1$, an ideal capacitor is as widely reported. When CPE has $\alpha = 1$, an ideal capacitor is described, while with $0.5 < \alpha < 1$, a distribution of relaxation times in the frequency space is represented [28–30]. Additionally, when considering the Nyquist plot, mainly at the intermediate frequency range, it seems that porous electrode behavior can also be characterized [30,31], which seems to be that corrosion behavior of the examined samples can be predicted by both planar and porous electrodes, as previously reported [28–31]. This is mainly when the blank samples and those containing up to 200 ppm are considered. This porous electrode effect seems to be minimized when the samples containing 400 and 600 ppm are examined. This seems to be associated with adsorbed compounds at the surface, and "porous/vacancies" are minimized or closed, as prescribed when surface cover values are analyzed, as well as the IE% values.

Corrosion is a complex process in terms of electrical currents where coatings or inhibitors are applied [32]. The complexity is caused by imperfectness of the coat due to different thickness levels and the existing pores. Therefore, this leads to more complex equivalent electrical circuits, which will include solution resistance ($R_{sol}$), double-layer capacitance ($C_{dl}$), as well as pore resistance ($R_{por}$). The current passes through the tinny openings and a thin layer of coating, which is in contact with the metal surface. This implies that the thinner the pores become, the more the current would be restricted—hence, pore resistance. The current that travels through these tiny pores creates interfacial impedance, which is connected in series with solution resistance and parallel with constant phase (CPE), Q1 in the EEC. Because of the thin film formation of the metal surface by the PBSLP, the charge transfer resistance from the metal to the bulk solution arises and this is connected in parallel with CPE, Q2 as shown in Figure 11.

The frequency response and phase-shift of dissolution of Fe from the mild steel surface in the absence and presence of PBSLP in 1 M hydrochloric acid are represented in Figure 10. It can be observed that the impedance of the system is high at low frequencies and decreases as the frequency is increased. Moreover, the phase angle increased with increasing PBSLP concentration. The deviation was observed with the blank solution; this could be attributed to the imperfections on the electrode surface. It could also be due to the impurities that were present with the electrode material, such as precipitates that

occur during forming. Overall, the PBSLP inhibitor worked to satisfaction, the corrosion rate of the mild steel was decreased with increasing PBSLP concentration, and the protective film was formed despite not being a perfect coating.

As for the selection of an appropriate EEC model, which can be applied to fit experimental data, the following conditions were used, as described by E. Barsoukov and J.R. Macdonald [33]. (1) The circuit diagram has an explicit meaning. (2) At each frequency point, real and imaginary impedance have minimum errors. (3) The resulting fitting has a minimum Chi-square value ($X^2$). The equivalent electrical circuit depends on minimum errors indicated by condition (2) if condition (1) and (3) are not met. In this work, conditions (1) and (3) were met; the Chi-square values obtained are shown in Table 2 and the EEC proposed represents solution resistance, interfacial impedance, and resistance due to blockage on the mild steel surface by the PBSLP inhibitor.

### 3.3. Surface Analysis

The protection of mild steel by PBSLP against corrosion was validated by observing mild steel coupons under the SEM after immersion in 1 M HCl. The immersion was done in the absence and the presence of PBSLP. Figure 12a–c shows the SEM micrographs of mild steel. As can be seen in Figure 12b, more surface damage is observed than in Figure 12c; this indicates that PBSLP protected mild steel against corrosion. Figure 12a,d were used as the references to the blank and 600 ppm PBSLP solutions. In this case, the PBSLP inhibitor blocked most of the active sides that were prone to corrosion, thereby reducing the corrosion rate of mild steel.

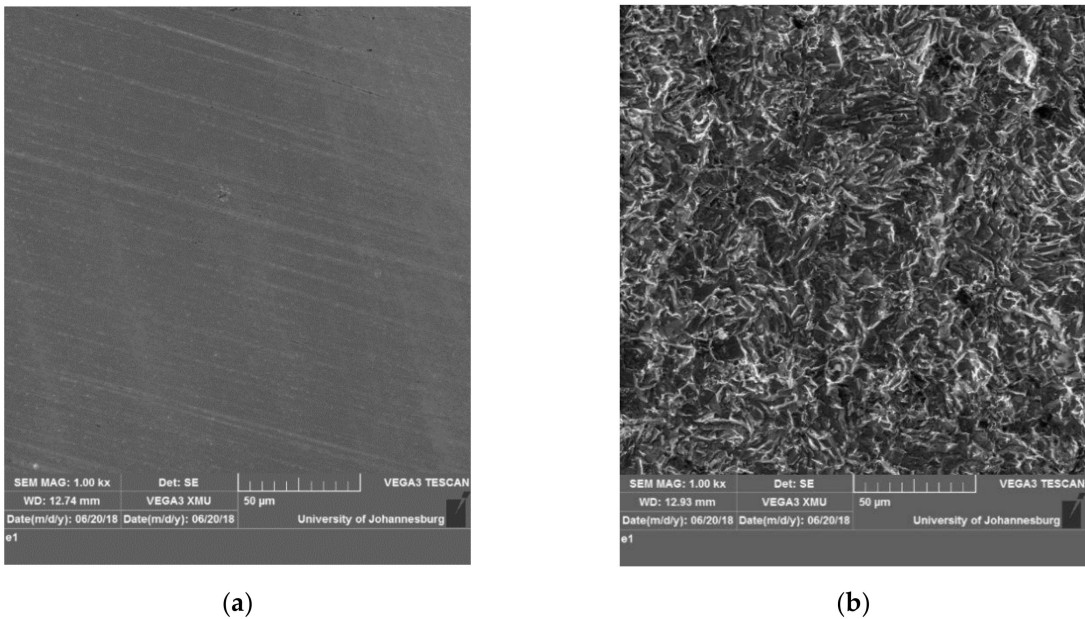

(**a**) (**b**)

**Figure 12.** *Cont.*

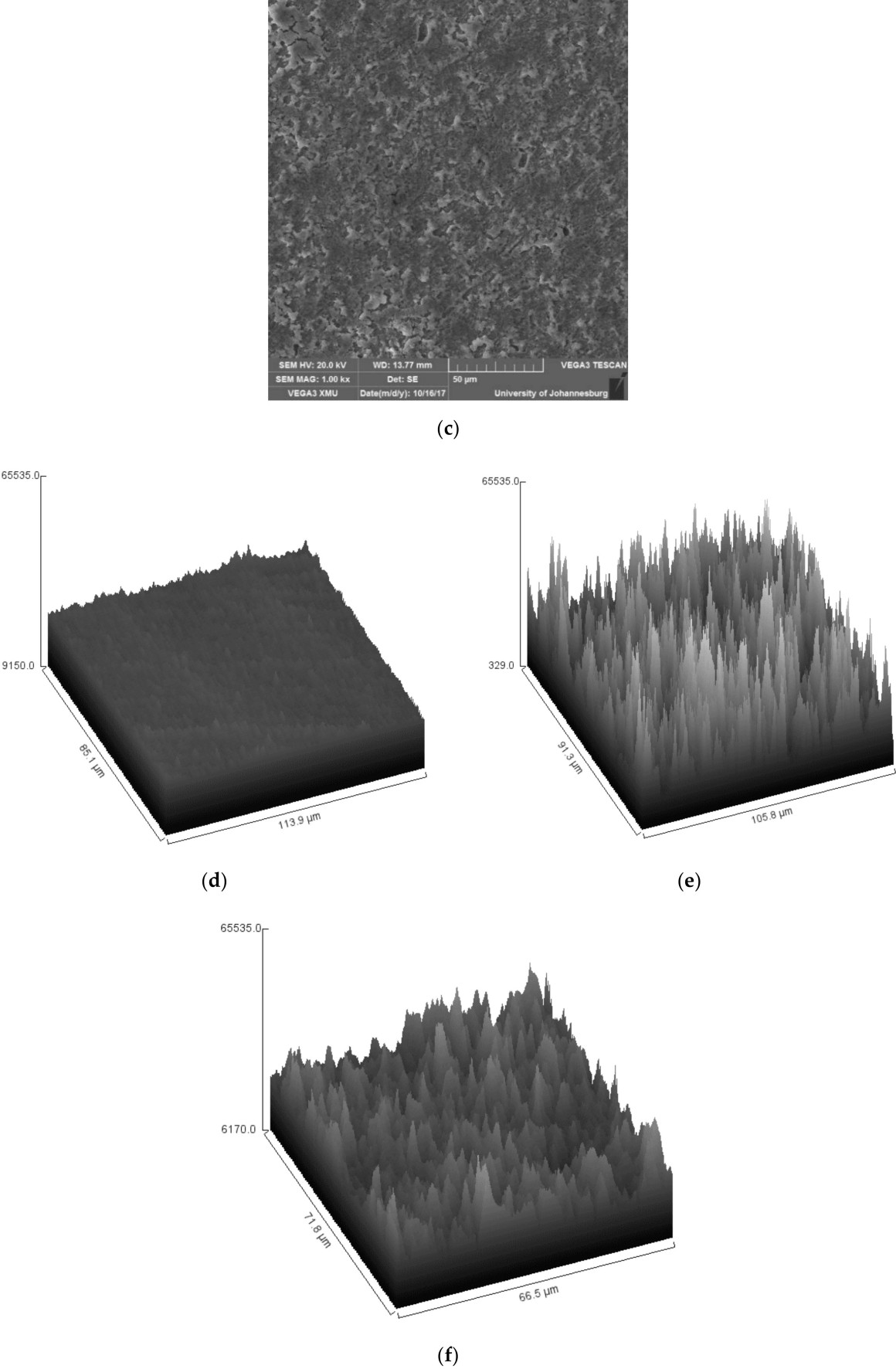

**Figure 12.** SEM micrograph of (**a**) unaffected, (**b**) blank solution (**c**) 600 ppm PBSLP and surface plot of (**d**) unaffected, (**e**) blank, and (**f**) 600 ppm PBSLP, respectively.

Topography characterization must be reliable, reproducible, and reveal important features. Scanning methods are widely used to characterize the surfaces of the material [34]. Among all the scanning methods, there is no insight obtained on the sharpness of surface topography. Therefore, in this study, the uncorroded and corroded mild steel surfaces were analyzed with Gwyddion 2.55 software to extract the vertical displacement of the action of HCl (corrosive media) from the SEM micrographs. The information obtained from Figure 12d–f provides the direct view sharpness of the peaks in all the samples. From the results, it can be observed that the coupons that were immersed in 600 ppm of PBSLP show protection with fewer sharp peaks as compared to the coupon in the blank solution, shown in Figure 12e,f, respectively.

Figure 13 shows the average roughness plot of mild steel coupons immersed in 1 M HCl, 600 ppm PBSLP, and the ground sample finished with 1200 grit paper. In Figure 13, the addition of PBSLP to 1 M HCl improved the roughness of the mild steel towards a smoother surface. The plots were obtained by Gwyddion software using SEM micrographs.

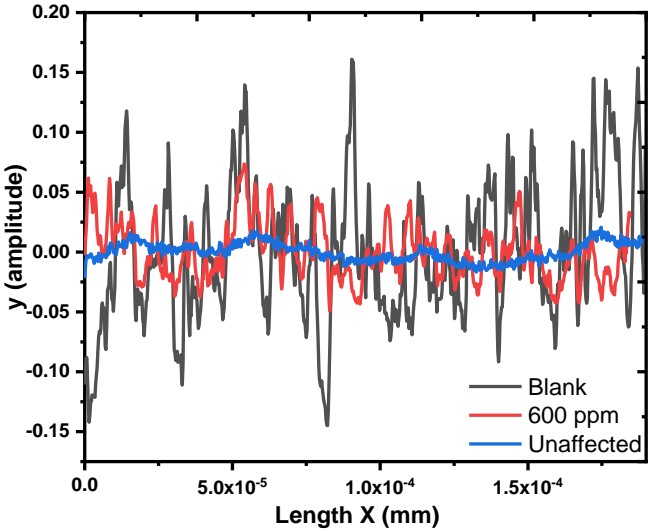

**Figure 13.** Average roughness plot of mild steel coupons immersed in the blank (black), 600 ppm PBSLP (red) solution, and ground coupon with 1200 grit paper (blue—unaffected).

Atomic force microscope (AFM) was also used to observe the surfaces of mild steel before and after exposure in 1 M hydrochloric acid in the absence and presence of PBSLP. Figure 14 shows the average roughness of the surface of mild steel coupons, Figure 14a shows the worst affected surface. A different observation is seen in Figure 14c because the presence of PBSLP in corrosive media offered protection to the sides that were prone to corrosion. There, PBSLP protected mild steel against corrosion by hydrochloric acid. Figure 14b was used as the reference to blank and 600 ppm PBSLP solutions.

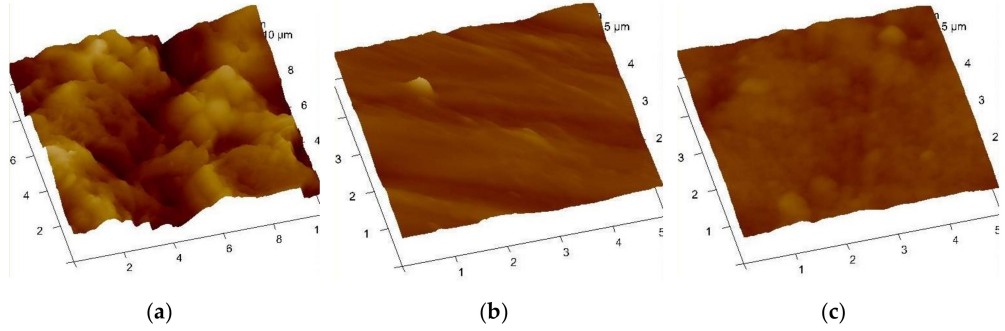

(a)            (b)            (c)

**Figure 14.** Atomic force microscope (AFM) roughness micrographs of (**a**) blank, (**b**) unaffected, and (**c**) 600 ppm PBSLP respectively.

### 3.4. Adsorption Studies

It has been documented that the inhibitive action of corrosion inhibitors is governed by the molecular structures of the inhibitors [35,36]. To determine the possible adsorption mode of PBSLP, the experimental data were tested with Langmuir and Freundlich adsorption isotherm models. The aim was to verify if the adsorption of PBSLP onto mild steel surface forms a monolayer and multilayer adsorption. The basic assumptions of Langmuir are: (i) the surface of the adsorbent is uniform—that is, all adsorption sides are equal; (ii) adsorbed molecules do not interact; (iii) all adsorption occurs through the same mechanism; and (iv) at maximum adsorption, only a monolayer is formed.

The fraction coverage values $\theta$ as the function of PBSLP concentration can be obtained from potentiodynamic polarization, as shown by Equation (8). Figure 15a shows the graphical representation of the Langmuir isotherm for the experimental data taken in 1 M HCl with different PBSLP concentrations. The data fit well with the Langmuir isotherm with the best fit value of $R^2$ of 0.998. The Freundlich adsorption isotherm model was also used to fit the data as $\theta$ vs. *log C*, shown in Figure 15b. The value of $1/n$ obtained from the slope was less than 1. For the multilayer adsorption, the value of $1/n$ must be between 1 and 10; then, adsorption would be favorable. In this case, Freundlich adsorption was not favorable [37]. The best fit value of $R^2$ of 0.962 was obtained. Therefore, from the $R^2$ values of the two models, it was concluded that adsorption of PBSLP onto the mild steel surface assumes monolayer adsorption [38]. The original data to this work can be found in the supplementary information.

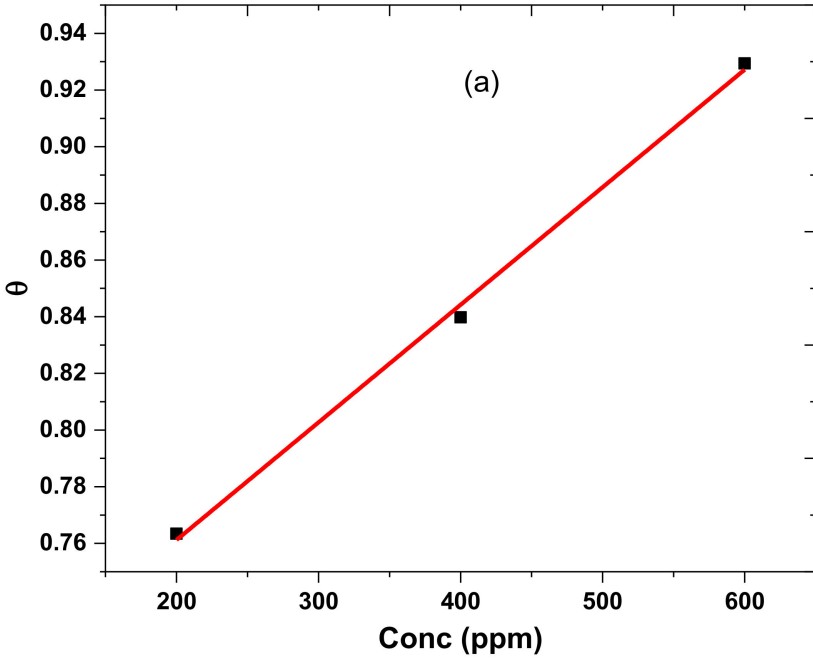

**Figure 15.** *Cont.*

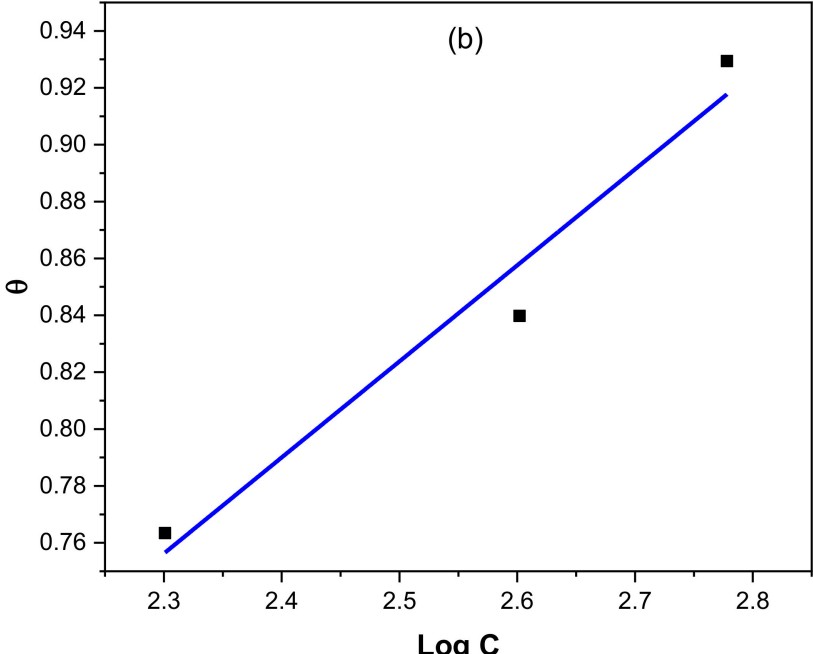

**Figure 15.** (**a**) Langmuir and (**b**) Freundlich adsorption isotherm model for adsorption of PBSLP onto the mild steel surface in 1 M HCl containing different inhibitor concentration.

**Supplementary Materials:** The supplementary materials were provided to the journal. The materials include figures and raw data that were used in the manuscript. The EC-Lab, Gwyddion, Vega 3, Inca, and Origin 8.5 pro were the softwares that were used to treat the data. The data is also available at https://data.mendeley.com/datasets/7ts5pndk3f/draft?a=fd9a0f35-9dc7-44f9-8060-41993229debf.

**Author Contributions:** E.M.M. was responsible for the supervision, project administration, and funding acquisition for this research work, while G.M.T. was responsible for conceptualization including all experimental and software validation of the work, writing, formal analysis, and editing of the manuscript. All authors have read and agreed to the published version of the manuscript.

**Funding:** This research was funded by National Resource Foundation (NRF), grant number: TTK150706123504.

**Acknowledgments:** The authors acknowledge the University of Johannesburg, specifically the Faculty of Engineering and Built Environment in the Department of Metallurgy for providing facilities and hosting of the authors.

**Conflicts of Interest:** The authors declare no conflict of interest.

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
