# Peer review of "Electrochemical and Surface Analytical Study on the Role of Poly(butylene-succinate)-l-proline during Corrosion of Mild Steel in 1 M HCl"

_chemistry, doi:10.3390/chemistry2040057_

Round 1

Reviewer 1 Report

The aim of this proposed manuscript has reasonable interesting. However, there are a great number of weaknesses, which induce to its REJECTION. The main weaknesses are follow indicated:

  1. English written should be meticulously be revised and improved;
  2. Reproducibility is rather and poorly detailed/discussed;
  3. EIS are merely and qualitatively discussed. A CNLS simulation should be provided;
  4. Polarization curves are carried out using a potential scan rate not commonly applied. No additional sentence is provided in order to explain this adopted potential scan rate;
  5. Nyquist and Bode-phase are confused and poorly plotted;
  6. Equivalent circuit is poorly discussed and depicted;
  7. Fig. 12 depicts Langmuir adsorption isotherm. Their corresponding parameters should be indicated inside a Table. Besides, at least another comparison, e.g. Freundlich, should also be included. Also, adsorption free energy parameters should be demonstrated; and
  8. Additionally, A great number of symbols are dismissed/equivocated written;

Author Response

  1. As recommended the introduction was improved.
  2. English had been revised by the authors.
  3. EIS had been revised, instead of CNLS simulator, the Z-fit tool on EC-lab software was used to analyse the data. 
  4. The scanning rate recommended by ASTM standard is 0.166 mV/s values ranging from 0.06 to 10 mV/s are found in the literature. Also, it is reported that there is no consensus choice of scanning parameters based on this information the scan 1.0 mV/s was adopted in this work.

Reviewer 2 Report

The synthetic conditions for the production of the “composite” is not disclosed. They claim it is similar to a previously published different material. Unless the process is identical, the method should be disclosed. Also, the form of the produced material (powder, film, liquid, etc) and the yield must be disclosed in a paper that claims to make a novel material.

I am not sure what is meant by a composite. What is the exact composite material. Is there a fiber and a resin? Nothing is described that suggests they are making a composite.

What is PBS? This abbreviation is used throughout and never defined.

The authors should give structures for the starting material and the composite. In particular, this is needed to support their FTIR spectral interpretation.

line 65 “starting with the course on until…” makes no sense.

line 73 and 77, range is “± 250 V” ‼ Assuming they meant mV, it is still incorrect (see Figure 5)

line 76 “KHz”, no such SI prefix.

No discussion of the FTIR spectral acquisition is given. How is the sample prepared and the spectra acquired. Sample dissolved or in a film? ATR or KBr pellet?  Nothing disclosed.

The Results section and the Discussion Section should be combined. The material in the results sections is repeated in the discussion section so it is entirely unnecessary

They use the abbreviation EDS and EDX for the same method. Pick one

For the FTIR, I suggest they use lines to indicate the peaks they discuss in the manuscript. It is difficult to follow their discussion otherwise.

The graphs should not have enclosing boxes as in Figures 2 to 12, but as in Figure 1

The text of the labels in Figure 3 and 9 should be white or light gray to make them visible

Figure 3 and from then on are incorrectly referenced in the text, they are off by 1

The SEM in figure 3? does not have any legend

The authors should use accepted abbreviations (s and h, not sec and hour)

Figure 6, The x and y axes of a Nyquist plot MUST be the same extent (here both from 0 to 40). Plotting with incorrect aspect ratios is misleading.

Figure 6, vertical axis label is incorrect (“-Iim”? imaginary current?)

Figure 8, what is this?  this is not even close to a circuit model, Where is Rs and Rct, what is Q2?. Q2 is only connected at one lead (if so, it would have no effect on the response). What is the function of the the word “Electrode”. Sloppy

Figure 7, Incorrect axis label for the phase

Figure 9, vertical axis has no units.

Figure 9, three different ways of referencing the labelled graphs (a) first, second (b), fourth -d.

Figure 10 vertical scale missing units

Figure 11, missing units

Table 1, Incorrect symbol for slope (β?)

Table 2, “act/” ?   “CPE/x10-3” what does that mean?

Line 157 what is meant by “distortion” not a normal term in IR. 

Line 157 1500 cm-1 is not the position I see in the spectrum. There is no peak at that wavenumber.

Line 177, How is the sample prepared for XRD, (bulk material, film on metal? 

Line 180 “a single high intense peak appeared to split and not distinct as compared…” No idea what this means

Line 182 “low intensity…complete dissolution” This is nonsense

Line 185, “The morphological appearance of molecules was…” No, you are not observing morphology of molecules. A film perhaps.

Line 187, There is no explanation of how this is true. “The SEM micrographs agree with the results obtained from XRD. In contrast, …” the authors just state things without any justification or explanation.

Line 188, “more than one phase” I see no evidence of that. Just a bunch of needle-like crystals

Line 221, “equation 1”, this should be equation 6

The EIS data is very sketchy. The authors do not show a valid model circuit. The circuit is missing elements they list in the table. The data does not appear to be replicated. The fit between the model and experiment is not show nor is any metric given to indicate the goodness of fit (χ2 or error). I have no confidence in any of this. The equation for the CPE should be given.

Line 263, “The closeness of value act to unity, the smooth the surface become. Therefore, from the results obtained is can be concluded with the addition of PBSLP inhibitor the surface became smoother.” This is complete nonsense.

Line 304, What is the Fourier Transform method they mention. No disclosure in the experimental section is provided and no other way to understand what this means. Terrible.

Line 309, “average roughness” from what method? AFM? no experimental detail given.

Line 314, Number without units.

Line 326, “surface coverage (?)”

Langmuir model. I don’t understand how a “composite” can be described as a monolayer. Monolayer of what exactly? Also, how is the surface coverage determined? Must be described.

Line 334, this link is broken, so I cannot see any supplementary material. If the authors have supplemental information, it must be described and referenced in the main MS.

Overall, this is a very poorly produced paper. The authors have done perhaps the correct experiments but they do not describe them properly or interpret them correctly. I cannot recommend publication in the current form. At a minimum, it requires major revision or perhaps rejection.

Author Response

  1. The introduction was improved as per recommendation. 
  2. The methods, results and discussions were revised and reporting improved.
  3. All the comments made were attempted.

Round 2

Reviewer 1 Report

REPORTS ON: chemistry-915427

When considering the first (original) version of the proposed manuscript, it has been decided to REJECT it. However, it seems that this revised version provides some improvements. Based on this, it seems that before its final publication, the follow modifications (MAJOR REVISION) should be provided:

  1. Concern to the adopted potential scan rate, into the revised version of the proposed manuscript, the follow sentence should be included:

“It is important to remember that potential scan rate has an important role in order to minimize the effects of distortion in Tafel slopes and corrosion current density analyses, as widely and previously reported [AA-DD]. However, based on these reports, the adopted 1mV/s can be considered without deleterious effects on Tafel extrapolation method in order to determine the corrosion current densities of the examined samples.”

[AA] Osório W.R., Freitas E.S., Garcia A, EIS and potentiodynamic polarization studies on immiscible monotectic Al–In alloys. Electrochimica Acta. 2013, 102: 436–445.

[BB] Osório W.R., Peixoto L.C., Moutinho D.J., Gomes L.G., Ferreira I.L., Garcia A., Corrosion resistance of directionally solidified Al–6Cu–1Si and Al–8Cu–3Si alloys castings. Materials and Design. 2011, 32: 3832-3837.

[CC] Zhang X.L., Jiang Zh.H., Yao Zh.P, Song Y., Wu Zh.D. Effects of scan rate on the potentiodynamic polarization curve obtained to determine the Tafel slopes and corrosion current density. Corrosion Science. 2009, 51: 581-587.

[DD] E. McCafferty. Validation of corrosion rates measured by Tafel extrapolation method. Corr. Scie 47 (2005) 3202-3215.

  1. In line 158, it is mentioned that “…a high frequency of 20 kHz to low frequency of 0.1 Hz...” . However, when Fig. 5 is observed, other frequency range can be seen. Please, it should be meticulously revised and solved.
  2. The values of “icorr” shown in Table 1 should also be depicted in Fig. 8 (Tafel plots). Tafel extrapolation should be indicated by using arrows inside Figure.
  3. Nyquist plots shown in Fig. 9 should be reworked in order to clarify the depressed semi arc. For this purpose, X and Y axes should be set in same scale.
  4. Additionally, in Fig. 9 (Nyquist) plots, at least one value of high frequency and another at low frequency should be depicted (Please see references #EE and #FF.
  1. Considering Langmuir and Freundlich adsorption isotherms, it is hardly suggested that into Table 1, the surface coverage (q) values and those corresponding with values of adsorption equilibrium constant and free energy of adsorption be included.
  2. Although at line 700, it is mentioned “minimum chi-square values”, these are not shown in Table 1. Since it has been mentioned that a CNSL (complex non-linear least squares) simulation was used, the each one corresponding chi-squared values should be shown, as also error ranges.
  3. English written between line 611 and 614 should be meticulously revised and improved.
  4. When discussing the elements in proposed equivalent circuit (between lines 685 and 687), it is hardly suggested that the follow articles and sentences be included, as follow:

“It is remarked that CPE characterizes the impedance of a phase element, which is described by CPE= [C(jw)n] -1, where C is the capacitance; j is the current (imaginary number: -10.5); w is the angular frequency and -1 < n < 1. When CPE attains n =1, an ideal capacitor is as widely reported [EE-GG]. When CPE has n = 1, an ideal capacitor is described, while 0.5 < n < 1, a distribution of relaxation times in the frequency space is represented [EE-GG].” Additionally, when considering the Nyquist plot, mainly at intermediate frequency range, it seems that porous electrode behavior can also be characterized [HH-II], which seems that corrosion behavior of the examined samples can be predicted by both planar and porous electrode, as previously reported [EE-FF; HH-II]. This mainly when the blank samples and that containing up to 200 ppm is considered. This porous electrode effect seems to be minimized when the samples containing 400 and 600 ppm are examined. This seems to be associated with adsorbed compounds at surface, and “porous/vacancies” are minimized or closed, as prescribed when surface cover values are analyzed, as well also the IE % values.

[EE] J.F.Q. Rodrigues, G.S. Padilha, A.D. Bortolozo, W.R. Osorio. Effect of sintering time on corrosion behavior of an Ag/Al/Nb/Ti/Zn alloy system. Journal of Alloys and Compounds 834 (2020) 155039.

[FF] L. M. Satizabal, D. Costa, P. B. Moraes, A. D. Bortolozo, W. R. Osório. Microstructural array and solute content affecting electrochemical behavior of Sn-Ag and Sn-Bi alloys compared with a traditional Sn-Pb alloy. Materials Chemistry and Physics 223 (2019) 410-425.

[GG] Bryan Hirschorn and A. Lasia (book: Electrochemical impedance spectroscopy and its applications, 2014).

[HH] B. Hirschorn, M. E. Orazem, B. Tribollet, V. Vivier, Isabelle Frateur and M. Musiani. Determination of effective capacitance and film thickness from constant-phase-element parameters. Electrochimica Acta, 55 (2010) 6218-6227.

[II] B. Hirschorn, M.E. Orazem, B. Tribollet, V. Vivier, I. Frateur and M. Musiani. Constant-Phase-Element Behavior Caused by Resistivity Distributions in Films. J. Electrochem. Soc., 157 (2010) C458-C463.

Author Response

Response to reviewer 1

  1. The authors appreciate the effort that the reviewer did take to review our manuscript to make it publishable. However, the authors found some of the recommendations not necessarily to be adopted in this manuscript. Example, comment number 3 on the reviewers report, having the values of Icorr and extrapolated lines within the is going to results in an untidy looking graph. The values are provided in table 1 and it is explained EC-Lab software was used to analyse the data therefore we found in the unnecessary to adopt that recommendation. Therefore, we did not find it necessary to show the values on the plot.
  2. On comment number 4, the Nyquist plot was reworked where the same scale was adopted on both x and y-axis to clearly show the depressed semi-circles as discussed. However, on comment number 5, labelling of the frequencies on the Nyquist plot while we explained in the method section that the frequency was scanned from high to low is unnecessary. The authors did not find it worthy to indicate frequency on the plot as recommended.
  3. Considering the adsorption isotherms, the aim was to establish which adsorption isotherm PBSLP follows. The adsorption constants and Gibbs energy of adsorption were not the main focus in this study therefore they were omitted.
  4. The tracking tool device on the word was utilized to indicate the changes made.
  5. The x and y scale on figure 9 was changed as recommended to show the depressed semicircle. 
  6. additional information provided by the reviewer was included in text together with the references.

Round 3

Reviewer 1 Report

Based on rebuttals provided by Authors, it is observed that improvements were adopted. With this, in my frank opinion, the paper deserves its final publication.

Best Regards;

 _ _ _ _